# Rapid Thermal Processing of Kesterite Thin Films

Maxim Ganchev [1], Stanka Spasova [1,*], Taavi Raadik [2], Arvo Mere [2] , Mare Altosaar [2] and Enn Mellikov [2]

[1] Central Laboratory of Solar Energy and New Energy Sources, Bulgarian Academy of Sciences, 72 Tzarigradsko Shaussee Blvd., 1784 Sofia, Bulgaria; mganchev@abv.bg

[2] Department of Materials Science, Tallinn University of Technology, Ehitajate tee 5, 19086 Tallinn, Estonia; taavi.raadik@taltech.ee (T.R.); arvo.mere@taltech.ee (A.M.); mare.altosaar@taltech.ee (M.A.); ennm@staff.ttu.ee (E.M.)

[*] Correspondence: perovskite.psc@gmail.com

**Abstract:** Multinary chalcogenides with Kesterite structure $Cu_2ZnSn(S,Se)_4$ (CZTSSe) are a prospective material base for the enhancement of the photovoltaics industry with abundant and environmentally friendly constituents and appropriate electro-physical properties for building highly efficient devices at a low cost with a short energy pay-back time. The actual record efficiency of 13.6%, which was reached recently, is far below the current isostructural chalcopyrite's solar cells efficiency of near 24%. The main problems for future improvements are the defects in and stability of the Kesterite absorber itself and recombination losses at interfaces at the buffer and back contacts. Here, we present an investigation into the rapid thermal annealing (RTA) of as-electrodeposited thin films of $Cu_2ZnSnS_4$ (CZTS). The treatment was carried out in a cold wall tubular reactor in dynamic conditions with variations in the temperature, speed and time of the specific elements of the process. The effect of annealing was investigated by X-ray diffractometry, Raman scattering and Scanning Electron Microscopy (SEM). The phase composition of the films depending on treatment conditions was analyzed, showing that, in a slow, prolonged, high-temperature process, the low-temperature binaries react completely and only Kesterite and ZnS are left. In addition, structural investigations by XRD have shown a gradual decrease in crystallite sizes when the temperature level and duration of the high-temperature segment increases, and respectively increase in the strain due to the formation of the phases in non-equilibrium conditions. However, when the speed of dynamic segments in the process decreases, both the crystallite size and strain of the Kesterite non-monotonically decrease. The grain sizes of Kesterite, presented by SEM investigations, have been shown to increase when the temperature and the duration increase, while the speed decreases, except at higher temperatures of near 750 °C. The set of experiments, following a scrupulous analysis of Raman data, were shown to have the potential to elucidate a way to ensure the fine manipulation of the substitutional Cu/Zn defects in the structure of CZTS thin films, considering the dependences of the ratios of $Q = I_{287}/I_{303}$ and $Q' = I_{338}/(I_{366} + I_{374})$ on the process variables. Qualitatively, it can be concluded that increases in the speed, duration and temperature of RTA lead to increases in the order of the structure, whereas, at higher temperatures of near 750 °C, these factors decrease.

**Keywords:** photovoltaics; multinary chalcogenides; thermal annealing; Kesterite; XRD

## 1. Introduction

The increased capacity of photovoltaics worldwide is could lead to the achievement of an ecological footprint of near-zero by the year 2030. Actually, silicon-based solar cell technology dominates the photovoltaics market, comprising near 95% of all installed modules [1]. Multinary chalcogenides are large class of compounds with special applications in photovoltaics. Beginning with chalcopyrite $CuInSe_2$, $Cu(In,Ga)Se_2$, and $CuInS_2$ (CIGS), a sustainable generation of compounds was established, successfully competing with silicon-based photovoltaic technologies [2,3] with record efficiencies of 23.4% for the Cd-free CIGS device [4] and 26.2% for CIGS-based solar cells with an electron back-reflector [5]. They

have several advantages, such as a sub-micrometer thickness, very high absorption coefficient near $10^5$ cm$^{-1}$ and tunable bandgap from 1.0 to 1.5 eV depending on the kind and ratio of chalcogens.

An interesting approach has been developed involving the formation of bifacial solar cells growing the structure with a Cu(In,Ga)Se$_2$ absorber on glass or a flexible substrate [6]. Recently, a record bifacial power conversion efficiency was demonstrated of near 20% and near 11% under frontal and rear illumination. The power generation density of the device was predicted to be comparable to the record obtained for the mono-facial option.

These factors encourage mass production but the scarcity of rare-earth elements in Ga limits the their potential application. In an attempt to overcome this obstacle, a new class of compounds of Kesterite was developed where the couple In and Ga is replaced with abundant elements Zn and Sn. Similar to Chalcopyrite (CIGSSe), the Kesterite Cu$_2$ZnSn(S,Se)$_4$ (CZTSSe) has a tetragonal crystal structure of Zinc Blende [7]. CZTSSe is a p-type semiconductor with a tunable bandgap between 1.0 and 1.5 eV, with direct transitions and a high absorption coefficient in the range of $10^4$–$10^5$ cm$^{-1}$ [8,9]. For several years, the record efficiency of the CZTSSe device, regardless of the deposition method was 12.6%, as reported in 2014 by IBM [10]. Despite significant efforts in recent years to improve the working characteristics of the CZTSSe solar cells, there are several notable issues that lead to weak performance parameters—deep level defects, a narrow phase stability region, and non-ideal device architecture. There is an approach to the modification of properties of the absorber material by doping or alloying with additional constituents as alkali dopants (Li, Na, K) or isoelectronic substitutions, but there are no reports of real improvements [11]; rather, effective management of the intrinsic defects was suggested to be the way to optimize the optoelectronic properties of the Kesterite-absorber materials.

An extensive and scrupulous analysis of strategies to improve the work characteristics of the Kesterite solar cells was recently presented [12]. In the abovementioned problems with bulk defects, special attention is paid to those in junctions with a buffer and back contact layers. The recently developed new record Kesterite solar cells with 13.6% efficiency is impressive [13]. An epitaxial Kesterite/CdS interface was reconstructed by the low-temperature-induced migration of Zn$^{2+}$ and Cd$^{2+}$, which were initially disordered by the solution treatment used for deposition of the buffer layer.

Low-temperature technology can enlarge the substrate material base by employing lightweight and flexible carriers [14]. For instance, there is the suggestion that flexible monograin CZTSSe solar cells can be built by combining a high-temperature synthesized absorber material in the structure [15]. In all cases, the optimal way to build thin-film Kesterite solar cells seems to be [12] the initial sintering process of the thin film absorber material, followed by a fast annealing process to avoid components' diffusion.

The aim of this work is to investigate the features of phase composition, films' morphology and a defect distribution in the results of the rapid thermal annealing (RTA) of electrodeposited CZTS thin precursor films. On the basis of the results obtained by XRD and Raman structural analysis, the phase composition is determined, containing the target CZTS and distribution of concomitants. In addition, assessment of the ratio of intensities of specific Raman signals can provide idea for the formation tendencies of some specific defects (as Cu/Zn substitutions) and approaches to the management of structural features by process parameters.

## 2. Materials and Methods

Substrate layers were electrochemically deposited in potentiostatic conditions using the classic three-electrode cell configuration. Working electrodes were $2 \times 1$ cm$^2$-sized tin-oxide-covered soda lime glasses positioned against platinum gauze and a saturated mercury sulfate (MSE) reference electrode (0.6151 V vs. standard hydrogen electrode). Electrolytes for the electrodeposition of Cu-Zn-Sn-S layers were 4 M KCNS aqueous solutions of 0.4 M sodium acetate buffer containing 9 mM of the chloride salts of Cu$^+$, Zn$^{2+}$, SnSO$_4$ and Na$_2$S$_2$O$_3$ in related ratios. The composition of the as-deposited Cu-Zn-Sn-S layers was

copper-poor and zinc-rich with sulfur deficiencies when compared to the stoichiometry. The substrate's influence on the composition of the layers was estimated to be negligible. The process design has been described in detail elsewhere [16]. Reactive annealing was performed in a cold walls' quartz tubular reactor under a flow of 5% $H_2S$ in $N_2$ at atmospheric pressure. Temperature profile was controlled in dynamic equilibrium by an IR Ulvac-RICO heating system supplied with a PC-driven controller. The system works under continuous cooling provided by the chiller-supplied circulation of working fluid at 10 °C and blowing with 0.6 MPa pressurized air. The heating was achieved by direct IR irradiation on the length of the reactor with xenon lamps. Under these conditions, at every segment, the process temperatures differed from set points with values of less than 0.1%. Below 220 °C, the ramp-down process temperature did not follow strictly the set point values, but it is believed that reactions here are faded. An example work temperature profile is shown in Figure 1.

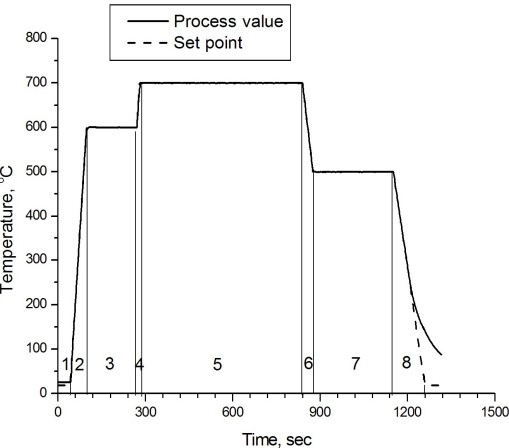

**Figure 1.** Temperature profile of rapid thermal processing of CZTS films.

The parameters for specific program configuration are summarized in Table 1. After the work cycle, the reactor volume was rinsed by $N_2$ stream for 5 min.

**Table 1.** Process parameters of rapid thermal annealing of thin-film CZTS.

| Probe | Base | Duration | Duration | Speed Fast | Speed Slow | T °C 750 °C | T °C 650 °C | T °C 600 °C |
|---|---|---|---|---|---|---|---|---|
| Sample | 1 | 2 | 3 | 4 | 5 | 6 | 7 | 8 |
| 1 Seg | 19 °C 10 s | 19 °C 10 s | 19 °C 10 s | 19 °C 10 s | 19 °C 10 s | 19 °C 10 s | 19 °C 10 s | 19 °C 10 s |
| 2 Seg | 600 °C 1 min | 600 °C 1 min | 600 °C 1 min | 600 °C 30 s | 600 °C 3 min | 600 °C 1 min | 600 °C 1 min | 600 °C 1 min |
| 3 Seg | 600 °C 3 min | 600 °C 3 min | 600 °C 3 min | 600 °C 3 min | 600 °C 3 min | 600 °C 3 min | 600 °C 3 min | 600 °C 3 min |
| 4 Seg | 700 °C 10 s | 700 °C 10 s | 700 °C 10 s | 700 °C 5 s | 700 °C 30 s | 750 °C 10 s | 650 °C 5 s | 600 °C 5 s |
| 5 Seg | 700 °C 15 min | 700 °C 10 min | 700 °C 5 min | 700 °C 15 min | 700 °C 15 min | 700 °C 15 min | 700 °C 15 min | 700 °C 15 min |
| 6 Seg | 500 °C 40 s | 500 °C 40 s | 500 °C 40 s | 500 °C 20 s | 500 °C 2 min | 500 °C 45 s | 500 °C 15 s | 500 °C 15 s |
| 7 Seg | 500 °C 5 min | 500 °C 5 min | 500 °C 5 min | 500 °C 5 min | 500 °C 5 min | 500 °C 5 min | 500 °C 5 min | 500 °C 5 min |
| 8 Seg | 19 °C 2 min | 19 °C 2 min | 19 °C 2 min | 19 °C 1 min | 19 °C 10 min | 19 °C 2 min | 19 °C 2 min | 19 °C 2 min |

Scanning electron microscopy and energy-dispersive X-ray analysis (EDAX) was performed on a Hitachi TM 1000 unit supplied with an X-ray source (Hitachi, Tokyo, Japan) and detector equipment at an accelerating voltage of 15.0 kV and acquisition time of 90 s. The X-ray diffraction (XRD) analysis was performed by a Rigaku Ultima IV diffractometer (Rigaku, Tokyo, Japan) with Cu-Kα radiation (λ = 1.5418 Å) at 40 kV accelerating voltage. The diffracted beam was scanned in steps by 0.01° for 2 s in an angular range from 10 to 80 degrees 2θ.

Qualitative phase analysis was performed on specialized software PDXL Rigaku's ICDD PDF2-phase research platform (Rigaku, Tokyo, Japan) using the specifications by International Center for Diffraction Data [17] and EDAX data for chemical composition. The suggested faze distribution by XRD is based on the detection of three or more main specific reflections of the phase compared with the actual database cards available at that time [17]. In addition, the broadening of the XRD peaks was used in an analysis of crystallites' size and internal strain. This was more popular than the Willimson–Hall method, which provides an idea of the deviations from the ideal crystalline lattice. The room-temperature (RT) micro-Raman spectra were recorded on a Horiba LabRam 800 high-resolution spectrometer (Horiba Ltd., Kyoto, Japan) equipped with a multichannel detector on backscattering regime. Light source was a red laser with a 633 nm wavelength focused on an at least 10 μm spot diameter, providing a spectral resolution of 0.5 cm$^{-1}$.

## 3. Results and Discussion

The reactive annealing is performed in compliance with both the suggested reaction paths and structural features in [18]. To conform with the proposed reaction sequence, the synthesis and structure formation of CZTS begin with the formation of binaries of $Cu_2S$ and $SnS_2$, followed by their interaction with the ternary $Cu_2SnS_3$ which, at higher temperatures, reacts with ZnS and completes as $Cu_2ZnSnS_4$ at temperatures higher than 600 °C. In this sense, the temperature profile configuration is directed to modifications of the area in which the direct formation of the quaternary CZTS takes place.

A detailed analysis of results for rapid thermal annealing consists of a scrupulous phase analysis and assessment of the influence of annealing process parameters on the crystal cell parameters of the target phase—Kesterite ($Cu_2ZnSnS_4$). In contrast to Stannite, where [19,20] fine differences exist between some closely disposed reflexes of $Cu_2ZnSnSe_4$ and either ZnSe or other binaries, and could become apparent through a more precise angle-resolved analysis, the case for Kesterite is rather more complicated [21] and an additional method such as Raman shift is essential for the correct evaluation of phase distribution.

Figure 2 shows XRD patterns of samples 1, 2 and 3 annealed with different durations (15 min, 10 min and 5 min, respectively) of the Segment 5 at 700 °C, according to Figure 1.

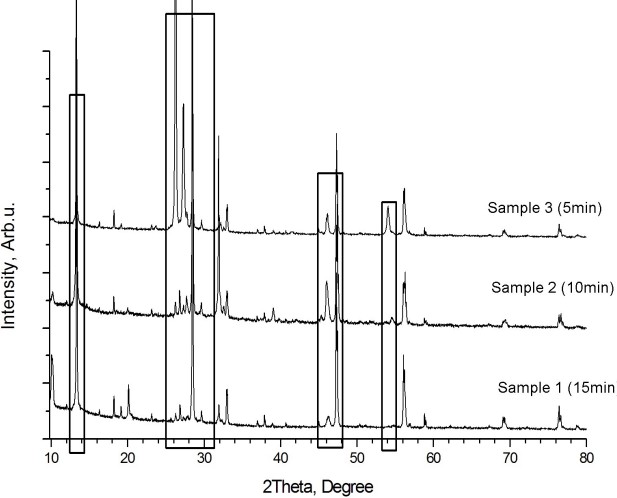

**Figure 2.** XRD pattern of samples annealed with different durations of Segment 5 (see Figure 1).

The patterns are dominated by characteristic reflexes for the Kesterite phase [1-01-075-4122] at 28.4° (1,1,2), 47.3° (2,2,0) and 56.1° (3,1,2) and its minorities at 16.32°, 18.23°, 23.13°, 29.6°, 32.9°, 36.9°, 37.9°, 40.7°, 44.9°, 56.9°, 58.8°, 67.3°, 69.1°, 76.4° and 78.8° in 2θ scale. Additional appreciable reflexes are detected near 10.2°, 13.3°, 26.2° 54° and 66.1° 2θ and confirm that the system is definitely not a monophase. The formal phase analysis performed through Ref. [17] and presented in Table 2 have specified the content of Sample 1: Kesterite, α-SnS [1:01-083-1758]; $Cu_7S_4$—[1:00-024-0061] and CuS—[1:03-065-3928].

**Table 2.** Phase composition according to XRD [17] and Raman shift analyses. ●—means registered phase; ◊—unregistered phase; Raman spectrum—−/+ (un)registered phase; ± not sure.

| Sample | Process | $Cu_2ZnSnS_4$ Kesterite-4122 | ZnS h | SnS | CuS Covellite | $Cu_7S_4$ Roxbyite | $Cu_2S$ digenite | ZnS h | ZnS h | ZnS(W) 0688h |
|---|---|---|---|---|---|---|---|---|---|---|
| 1 | Base 15 min, 700 °C | ●+ | ●2201+ | ◊ | ◊ | ◊ | ◊− | ●2424 | ●2195 | ◊ |
| 2 | Time—10 min | ●+ | ●2347 | ●+ | ● | ● | ◊− | ◊ | ◊ | ● |
| 3 | Time—5 min | ●+ | ◊ | ●+ | ● | ● | ◊− | ◊ | ◊ | ◊ |
| 4 | Speed—Fast | ●+ | ●4998+ | ● | ◊ | ◊ | ◊± | ◊ | ◊ | ◊ |
| 5 | Speed—Slow | ●+ | ●6022+ | ◊ | ◊ | ◊ | ◊ | ◊ | ◊ | ◊ |
| 6 | Temp—750 °C | ●+ | ●2140 | ◊− | ◊ | ◊ | ●9133/− | ◊ | ◊ | ◊ |
| 7 | Temp—650 °C | ●+ | ●4989 | ●+ | ◊ | ◊ | ◊± | ◊ | ◊ | ◊ |
| 8 | Temp—600 °C | ●+ | ●6009 | ●+ | ◊ | ◊ | ◊ | ◊ | ◊ | ◊ |
| 1 | Base—700 °C | ●+ | ●2201+ | ◊ | ◊ | ◊ | ◊± | ●2424 | ●2195 | ◊ |

As mentioned above, the main phases, concomitant with the Kesterite dispose reflexes, are quite close to those for the main phase and a more precise analysis could reveal the fine structure of the films. Figure 3 presents part of the pattern of Figure 2 in the vicinity of 13–14° 2θ. The picture presents the fine disposition of reflexes, which allowed for the one for ZnS at 13.35° 2θ (Sample 3) to be distinguished from the other at 13.28° 2θ (Sample 2) belonging to $Cu_7S_4$.

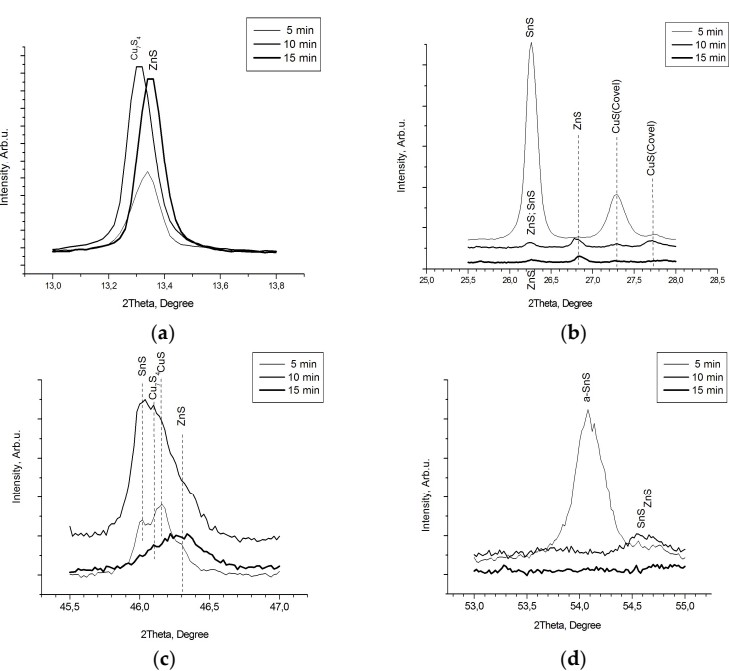

**Figure 3.** Pattern sections from Figure 2. (**a**) Pattern at 13.0–14.0° 2θ. (**b**) Pattern at 25.5°–28.0° 2θ. (**c**) Disposed pattern between angles 45.5°–47.0°. (**d**) Pattern at 54°.

Figure 3b presents pattern from Figure 2 in the interval 25.5°–28.0° 2θ. In accordance with the performed phase recognition, the reflex at 26.24° for Sample 1 should be assumed to come from SnS only; for Sample 3, the small reflex at the same angle should be assumed for ZnS only, but for Sample 2, the reflex should be consistent with ZnS and SnS.

At 26.82°, 2θ are the disposed reflexes of only Samples 2 and 3, recognized as being from ZnS, whereas, at 27.28° and 27.75° 2θ, single signals can be seen from CuS for Samples 1 and 2. Further, in Figure 3c, patterns between angles 45.5°–47.0° 2θ are shown. Here, the large signal from Sample 3 is due to ZnS. For Samples 1 and 2, the reflex at 46.02° is proven for SnS, whereas those at 46.11° and 46.18° are caused by $Cu_7S_4$ and CuS, respectively.

Figure 3d shows the field near 54° 2θ in detail. The reflex at 54.05° for Sample 1 comes from α-SnS [1:01-083-1758], whereas, for Sample 2, the reflex at 54.5° is due to SnS [1:01-073-1859] accompanied by Wurtzite [1:00-012-0688].

Figure 4a shows XRD patterns for Sample 4 (fast) and Sample 5 (slow), which, in comparison with Sample 1(base), present an RTA configuration with different rising and decreasing temperature speeds for the dynamic segments No. 2, No. 4, No. 6 and No. 8 (see Figure 1). The right values of rising speed are 3.3 °C/s, 10 °C/s and 20 °C/s, whereas at decreasing steps they are 1.33 °C/s, 5 °C/s and 10 °C/s, respectively.

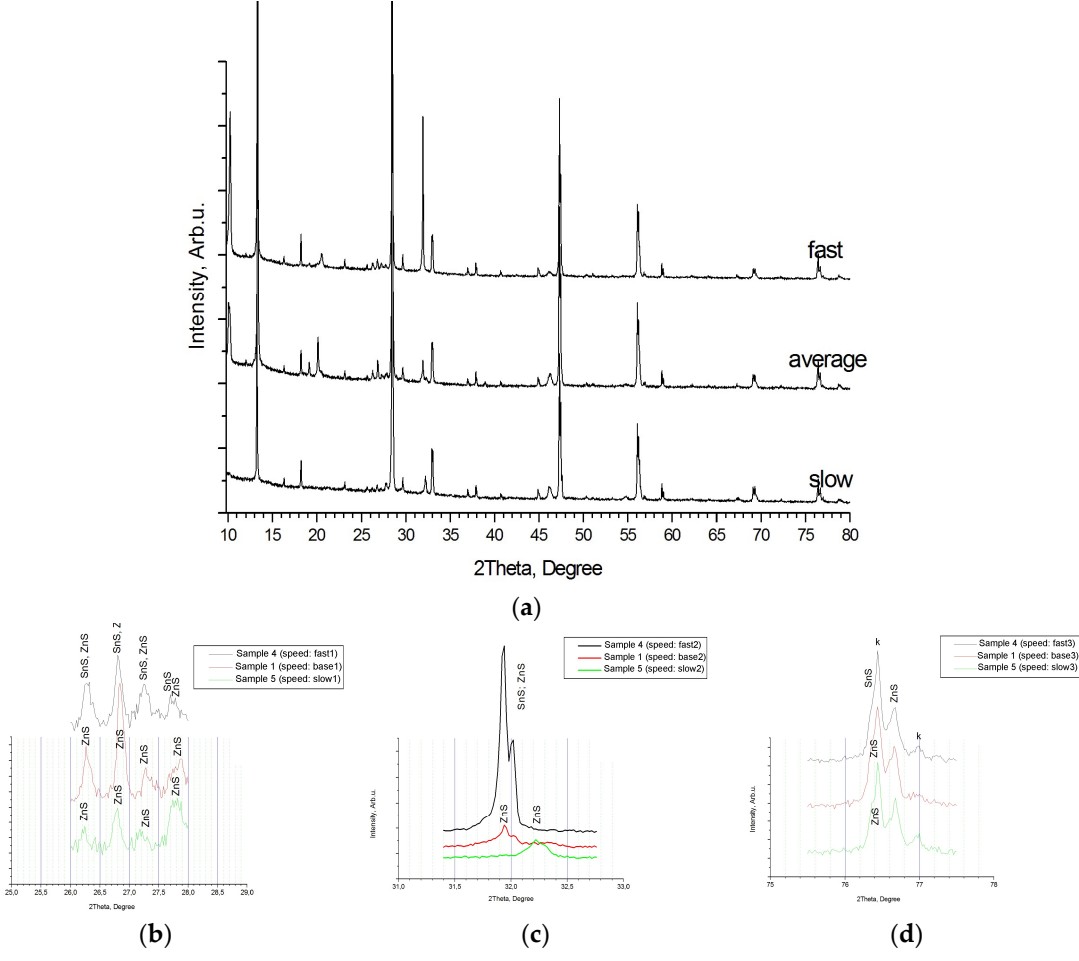

**Figure 4.** XRD patterns of samples annealed at different speeds (see Figure 1—Segments No.: 2, 4, 6 and 8). (**a**) Superimposed diagram of all speeds. (**b**) Patterns in vicinity of 26°–28° 2θ. (**c**) Patterns in vicinity of 31.5°–32.5°. (**d**) Patterns in vicinity of 76°–77°.

According to Ref. [22], the phase distribution of the samples is summarized in Table 2. Sample 1, annealed at faster temperature increases, contains mainly Kesterite. A pair of SnS phases—orthorhombic [1:01-072-8499] and [1:00-001-0984]—are registered, as well as

a Hexagonal ZnS [1:01-074-4998]. For Sample 5, annealed at a slow rate, Kesterite and Hexagonal ZnS [1:01-075-6022] are found. As previously mentioned, the average rate of Sample 1 contained Kesterite and Hexagonal zinc sulphides. A comparative phase analysis shows the Kesterite and zinc sulphides as common structures in the three samples, whereas Sample 4 contains two forms of Herzenbergite [1:01-072-8499 and 00-001-0984]. They have a similar layered orthorhombic structure and differ in their orientation. Obviously, both of them are a result of non-equilibrium interactions in the chain of the formation of CZTS caused by the high-speed changes in the temperature. Apparently, under softer conditions, such as a longer reaction time (previous case) or slower dynamic process parameters, in accordance with Ref. [23], this intermediate further interacts in a related scheme. Figure 4b shows the XRD pattern of the samples in the vicinity from 26° to 28° 2θ. Reflexes near 26.2° are assumed, for all samples, to be from ZnS, but for Sample 4 they appears to be in convolution with a signal from SnS. In the same way, the reflexes at 26.8° and at 27.3° 2θ can be interpreted. Near 27.75° 2θ, the disposition is quite similar but the complex reflex for Sample 4 looks well-divided for both SnS and ZnS. Figure 4c presents the interval 31.5°–32.5° 2θ. Sample 5 disposes an individual reflex at 32.23°, whereas the other Samples, 1 and 4, present a similar twin reflex at 31.94° and 32.02° 2θ. Reference [22] spelt out the twin for Sample 1 as coming from ZnS only, while, for Sample 4, the pair of ZnS and SnS is assumed. This could be explained by the obvious difference in the height (intensity) of the reflexes. Between 76° and 77° 2θ, Figure 4d shows disposed reflexes at 74.4° and 77°, assumed to be Kesterite. ZnS is presented here, with a reflex at 76.68° and a shoulder at the first Kesterite reflex. For Sample 4, a twin shoulder is distinguished, caused by both ZnS and SnS. The Raman spectra presented in Figure 5 are in good agreement with the supposed phase compositions of Samples 4, 5 and 1. In all samples, the Kesterite is well-defined, with the majority of its characteristic vibrations occurring at 338 cm$^{-1}$, 289 cm$^{-1}$, 251 cm$^{-1}$, 98 cm$^{-1}$ and 375 cm$^{-1}$. For all samples, a shoulder at 352 cm$^{-1}$ is seen, near to the most intensive signal for Kesterite, attributed to ZnS. For Sample 4, a resonant vibration at 164 cm$^{-1}$ is registered, attributed to SnS, in confirmation with the established XRD phase distribution.

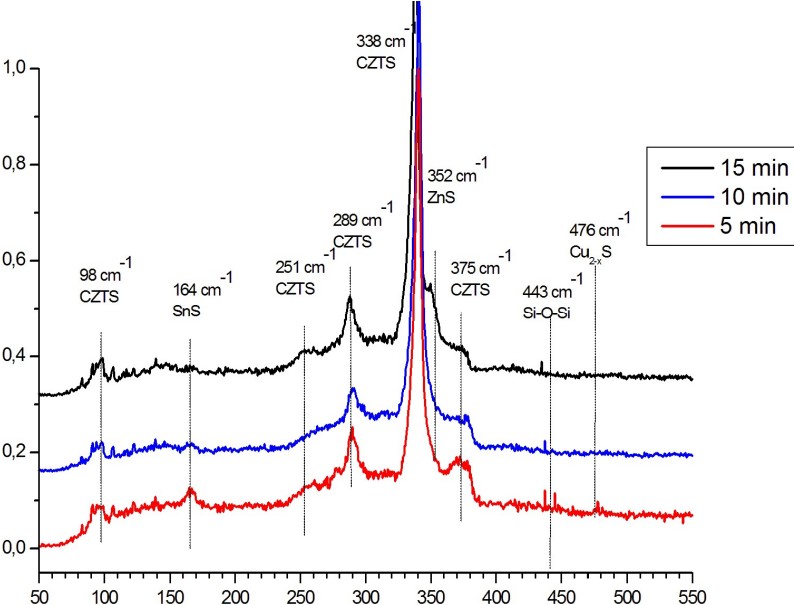

**Figure 5.** Raman shift—duration.

Figure 5 presents the Raman spectra of the same films. In accordance with [23–25], it can be concluded that the spectra are again dominant for Kesterite signals at 338, 287, 252, 374 and 97 cm$^{-1}$. After [26,27] the disposition of the shoulder near 350 cm$^{-1}$, in conjunction with signals at 98 cm$^{-1}$ and possibly near 160 cm$^{-1}$, the distribution of ZnS can

be revealed. Similarly, the shift at 164 cm$^{-1}$ and the shoulder near 311 cm$^{-1}$ confirm the presence of SnS [8,22,25,28,29]. Vibration at 476 cm$^{-1}$ is characteristic of the Cu-S bond in nonstoichiometric unsaturated copper sulphides [27]. This comparative analysis confirms this possible suggested phase distribution, but full identification requires a more detailed view of the XRD patterns.

Figure 6 presents the XRD pattern of Samples 6, 7 and 8, annealed at different temperatures (level of Segment 5, Figure 1) of 600 °C, 650 °C and 750 °C, respectively, which, together with Sample 1 (700 °C for level of Segment 5) are the process set configuration for investigations of the influence of the temperature. The patterns present a well-crystallized Kesterite in all samples, accompanied by different ZnS phases in almost the same structure [1:01-073-6009, 01-074-4989, 01-089-2201, 01-089-2424, 01-089-2195, 01-089-2140].

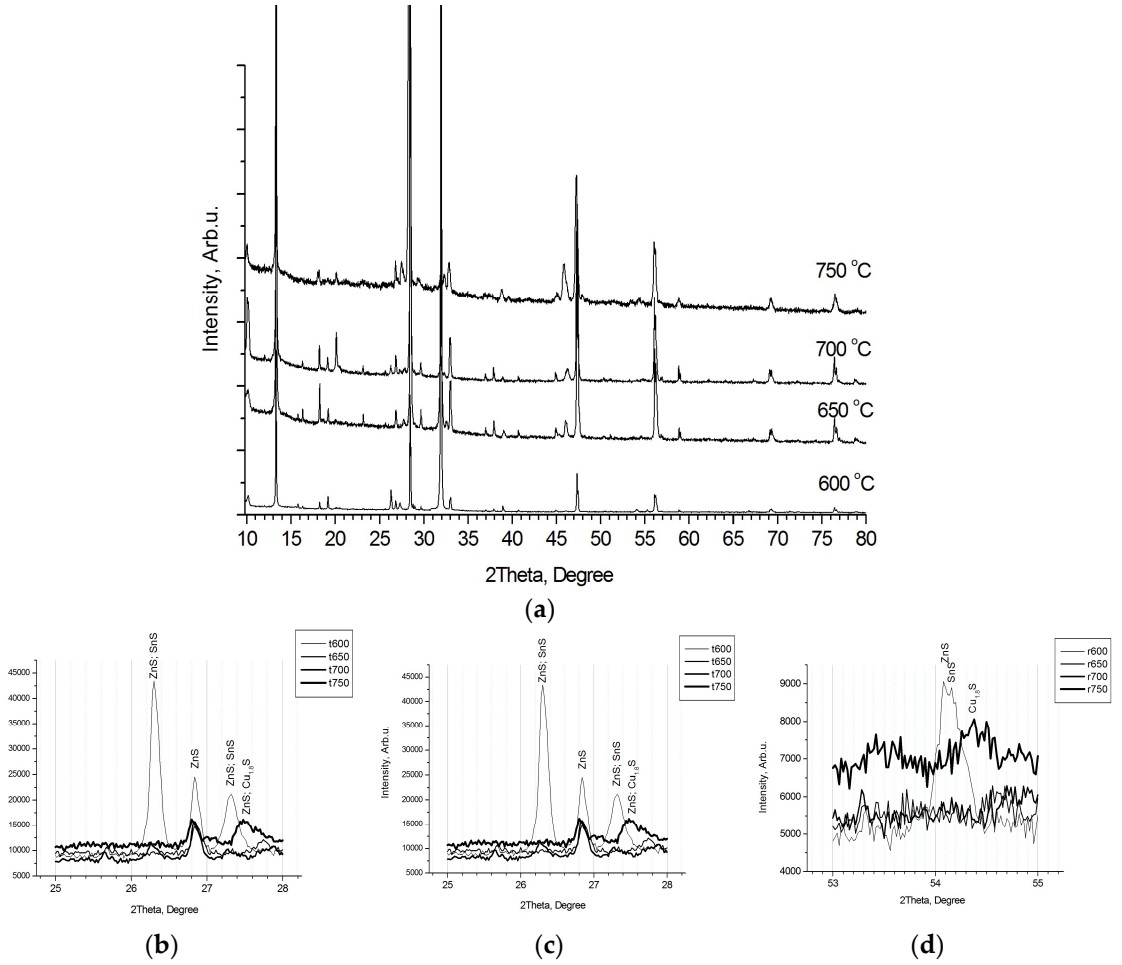

**Figure 6.** XRD patterns of Segment 5, RT-processed at different temperatures (see Figure 1). (**a**) Superposition at different temperatures. (**b**) Pattern at 26°–28°. (**c**) 45.5°–47°. (**d**) 53°–55°.

For low-temperature Samples 6 and 7, annealed at 600 °C and 650 °C, respectively, hexagonal tin sulphides are registered, while for high-temperature Sample 8 (750 °C), Digenite [1:01-070-9133] is attributed. A detailed XRD analysis in the interval 26°–28° is shown in Figure 6b. Sample 6 disposed three reflexes (at 26.30°, 26.83° and 27.32° 2θ), attributed to ZnS/SnS, ZnS and ZnS/SnS, respectively. The reflex at 26.83° is common for all samples, whereas Sample 8 had a peak at 27.5° 2θ, identified as ZnS that may be Cu$_{1.8}$S. Figure 6c shows the interval of angles between 45.5° and 47°. Samples 5 and 6 (600 and 650 °C) showed reflexes at 46.08°, recognized as being caused by SnS. Between 46.2° and 46.4°, a large reflex is seen for Sample 1, attributed to ZnS. Sample 8 showed a peak between 45.80° and 45.82° 2θ, attributable to ZnS, which may be Digenite, and a shoulder

at 46.08°, recognized to be caused by SnS and which may be Digenite. Further, in Figure 6d at 53°–55° 2θ, Sample 5 (600 °C) revealed a twin at 54.09° and 54.16°, whereas for Sample 8 (750 °C), a signal from Digenite was recognized at 54.38°.

Figure 7 presents a set of patterns for Raman shifts in the same samples. Raman spectra showed the presence of Kesterite, ZnS and SnS for Sample 5 and Sample 6, and Kesterite and ZnS for Sample 1 and Sample 8, but there was no confirmation of an Cu-S bond. In sequence, the phase composition features recognized from XRD [22] analysis are confirmed by Raman shift, with the exception of the phase $Cu_{1.8}S$. XRD reflexes were not uniquely defined and the absence of specific Raman resonant vibrations provides reasons to consider this phase as non-existent.

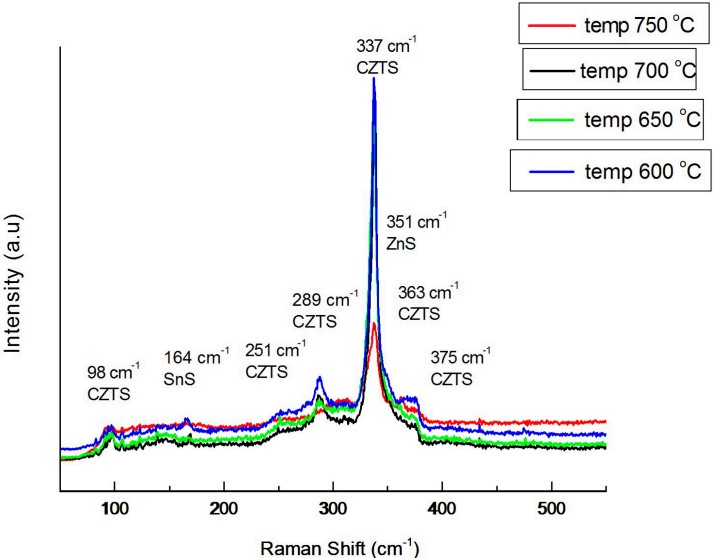

**Figure 7.** Raman shift for samples annealed at different speeds.

For Sample 2 (10 min), the same phases were recorded, but SnS is in another structure configuration [1:01-073-1859], in addition to ZnS [1:01-089-2347 and 00-012-0688]. Sample 3 (15 min) is common for all other RTA configurations and consists of only Kesterite and Hexagonal ZnS [1:01-089-2201, 01-089-2424 and 01-089-2195].

Figure 8 presents the dependences of crystallite size and lattice strain, calculated by the Williamson–Hall method [17] on process parameters for sets used to investigate the influence of the duration of Segment 5 (Samples 1, 2 and 3), speed of temperature changes in dynamic segments 2, 4, 6 and 8 (Samples 4, 1 and 5), and temperature level of Segment 5 (Samples 6, 7, 1 and 8) at 600, 650, 700 and 750 °C, respectively.

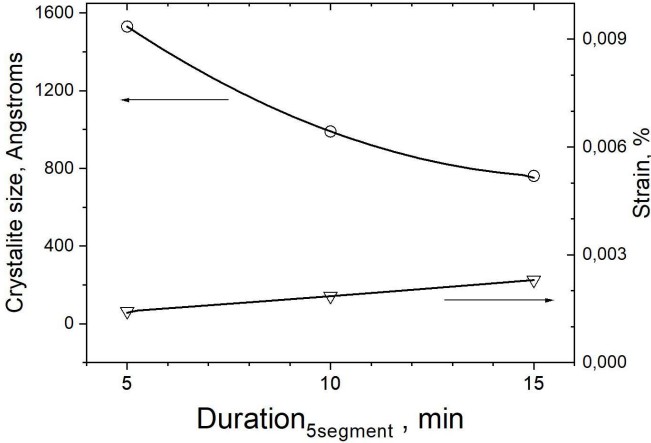

**Figure 8.** Crystallites size and strain of Kesterite formed by RTA at different durations.

As can be seen, with the increase in duration of Segment 5 from 5 to 15 min, the crystallite size decreases from 1530 Å to near 760 Å, whereas the strain of the cell rises from 0.00143% to 0.00230%. The same trend is seen for the temperature, where the crystallite size drops from 1193 Å for 600 °C to 609 Å for 750 °C, respectively, and the strain rises from 0.00133% to 0.0036%.

Regarding the dependence of crystallite size and strain on the speed of changes in temperature, the general trend is the same, but the curves are not monotonous, as can be seen in Figure 9. The crystallite size is 1521 Å for the fast process and decreases to 1010 Å for the slowest process, while, at average speed, the size is as low as 761 Å. The strain drops from 0.00139% to 0.00026% while, at average speed, the maximum is 0.0023%.

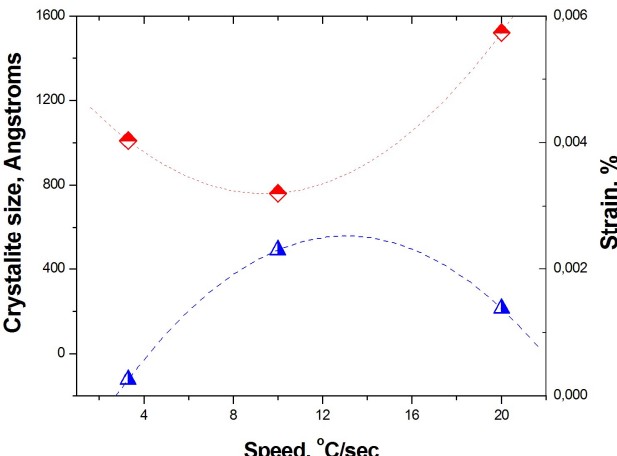

**Figure 9.** Crystallite size and strain of Kesterite formed by RTA at different speeds.

Regarding the influence of temperature, as shown in Figure 10, in comparison with the data presented in [29], a remarkable difference can be seen. While the cell parameters for 700 °C at [29] are a = 5.6177 Å and c = 11.2232 Å, respectively, in our case they are a = 5.4262 Å and c = 10.8519 Å. This is the reason that a difference in the cell volume is registered at 600 °C from 335 Å$^3$ [29], whereas cell volume in our case is near 320 Å$^3$.

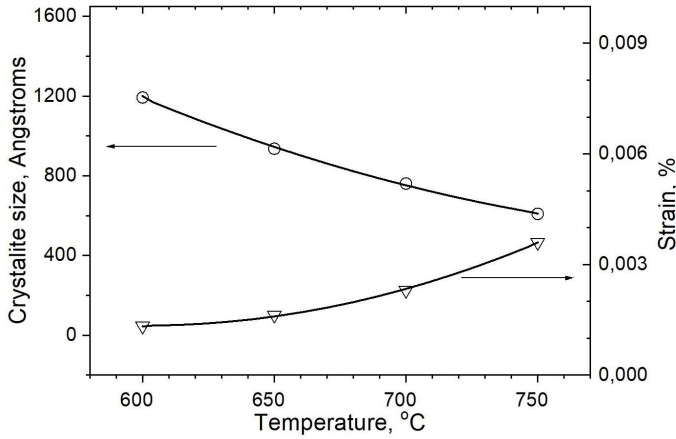

**Figure 10.** Crystallite size and strain for Kesterite phase formed by RTA at different temperatures.

Generally, it can be concluded that a slower and longer process at higher temperatures for copper-poor, zinc-rich, near-stoichiometric substrates leads to a preferable phase composition of Kesterite and zinc sulphides, whereas the cell strain increases with increasing temperature and an increase in the duration of Segment 5, but decreases when the speed of the process decreases. The results are in agreement with Ref. [8] and Ref. [30], but differ from the newer works in some details [29,31]. For example, the difference in cell parameters [29] at 700 °C is easy to explain when considering the dynamics of the

process—in Ref. [29], the process nears equilibrium whereas, in our case, a time-resolved (faster) process is shown, which, in slower stages, coincides with previous results [29–31].

Figure 11 presents SEM micrographs of samples, annealed in different duration.

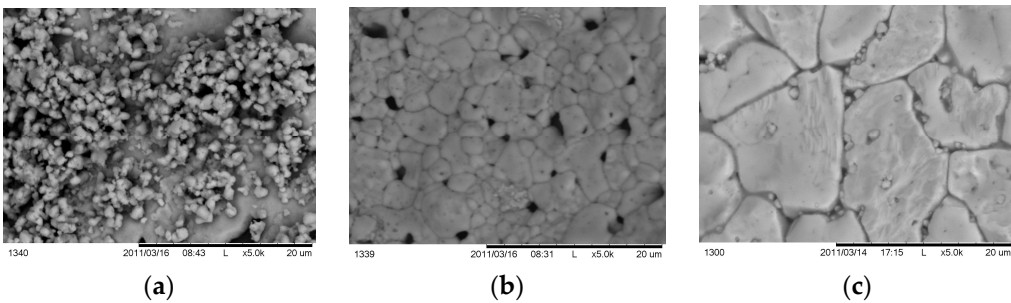

**Figure 11.** SEM footprints of samples, annealed at different times (Segment 5, Figure 1): (**a**) 5 min; (**b**) 10 min; (**c**) 15 min.

All scale bars in the figures are equal to 20 μm. For a short annealing time, a small part of the as-deposited layer crystallized on the substrate area, leaving unreacted mass on the top surface. For 10 min (Figure 11b), a pin-holes layer is formed with grain sizes between 1 and 4 μm, whereas at 15 min (Figure 11c), the grains are much larger, with a size 4–5 μm.

Figure 12 presents morphology issues for samples annealed at different speeds. The grain sizes enlarge from 1–2 μm for fast (Figure 12a) annealing, through 4–5 μm for average annealing (Figure 12b), and reaching 8–10 μm for samples annealed using a slow process (Figure 12c).

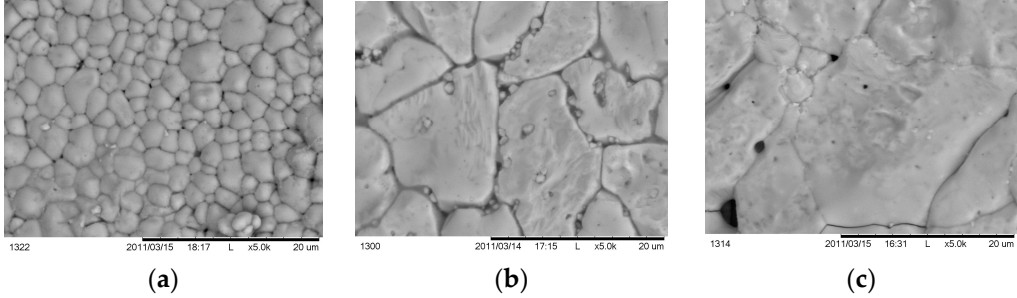

**Figure 12.** SEM fingerprints of samples RTA annealed at different speeds: (**a**) fast process, (**b**) average speed; (**c**) slow process.

Figure 13 shows the morphology footprints for samples processed at different temperatures. Figure 13a presents a CZTS layer with single independent grains with sizes near 1–2 μm. At a temperature of 650 °C, the surface concentration of the grains increased, and pinholes are seen. At 700 °C, the grain sizes reach a maximum of 4–5 μm while, at a higher temperature if 750 °C, the sizes grew smaller with heterogeneous residuals on the grain borders.

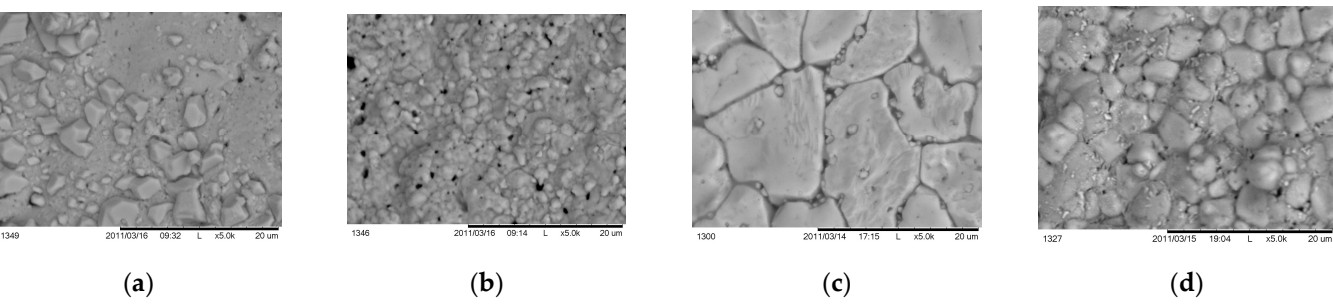

**Figure 13.** SEM micrographs for samples annealed at different temperatures: (**a**) 600 °C; (**b**) 650 °C; (**c**) 700 °C and (**d**) 750 °C.

SEM observations a gave slightly different impression of the crystallite size from the calculated values. As shown in Figures 8–10, following XRD analysis, crystallites sizes ranged from 0.05 to 0.15 μm whereas SEM revealed sizes near 1–8 μm.

Here, we can conclude that the tendencies of changes in crystallite size, derived from XRD data obtained using Williamson–Hall method [17], differ by an order of magnitude and, in addition, are the opposite of the tendencies of changes in grain sizes derived from SEM. In fact, the Williamson–Hall method revealed that short-ordered crystal unit Kesterite sizes are quite different from the visible grain structure of the layers.

As was reported elsewhere [32,33], the Raman spectroscopy could be used for characterizing Cu/Zn disorders and to distinguish the origin of the substitution in slow-cooled Kesterite thin films. The assessment of the level of disorder is very important for the device output. Scrupulous comparative analysis with Nuclear Magnetic Resonance (NMR) data revealed the ratio of specific Raman shifts at 287 cm$^{-1}$ and 303 cm$^{-1}$, $Q = I_{287}/I_{303}$, and additionally introduced a shift between an intensity of 338 cm$^{-1}$ and the sum of intensities at 366 and 374 cm$^{-1}$, $Q' = I_{338}/(I_{366} + I_{374})$, which is very sensitive to the Cu/Zn disorder. Here, a rough idea of the dependencies in time-resolved RTA processes is presented.

Figure 14a presents the dependencies of the Raman peak ratios, with intensities of $Q = I_{287}/I_{303}$ and $Q' = I_{338}/(I_{366} + I_{374})$, with increases in the duration of the high-temperature annealing step in this process. It can be seen that both Q and Q' are higher than 1 in all cases, which is characteristic of an ordered material [32] and increase with increases in the duration. The orders-of-magnitude of increases in the Q are quite small—from 1.39 through 1.5, up to 1.86—whereas the Q' is quite higher and increases by almost twofold, from 2.9 up to 4.7. The same trend is observed regarding the dependence of Q and Q' on the temperature (level of segment 5), but at the higher level of 750 °C, both Q and Q' drop sharply. This effect could be interpreted as an increase in the disorder of the Kesterite structure. The increase in the speed of the increase and decrease in temperature influence the Q and Q' in the opposite way. This effect can be explained by increases in the order of the structure at high speeds in the dynamic stages, bringing the process closer to the case of quenching in the experiments in [33].

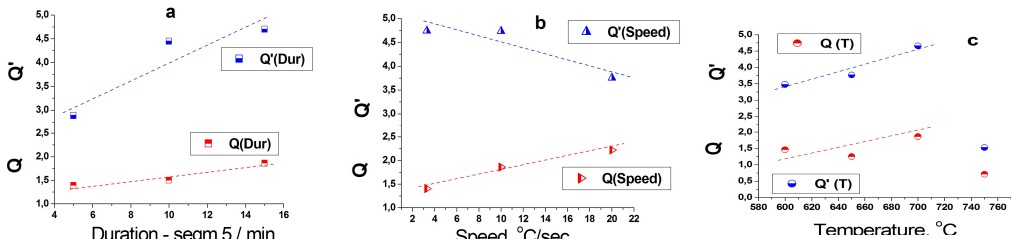

**Figure 14.** Dependences of Raman peak ratios $Q = I_{287}/I_{303}$ and $Q' = I_{338}/(I_{366} + I_{374})$ on the RTA process parameters: (**a**)—duration of the segment 5; (**b**)—speed of the change in temperature in the dynamic steps—segments 2, 4 and 6; and (**c**)—temperature of segment 4 (see Table 1).

## 4. Conclusions

A time-resolved annealing process is performed for CZTS thin films. The peculiarities of the phase composition and structure parameters of the formed thin Kesterite layers regarding the dependence of these process parameters (such as time, speed and temperature) are analysed. It is shown that the phase composition of the layers depends on the time and speed of the stage of Kesterite formation. Soon, only unreacted binary residuals, such as copper sulphides and tin sulphides, are left. The dependence is the same for a fast process where the dynamic change in temperature does not allow for binary reaction precursors to react up to the quaternary compound. If the reaction is slow and long enough (at least 15 min in our case), low-temperature binary precursors, according to Schurr et al. [18], react fully and only Kesterite and the high-temperature ZnS are left. The situation is the same regarding the temperature of the CZTS formation process. At low temperatures, unreacted binary copper and tin sulphides are registered, whereas at

high temperatures (700–750 °C), well-crystallised Kesterite is formed with ZnS immersions, which is a favourable phase composition regarding the photovoltaic properties of absorber films. However, the time-dependent properties of the Kesterite are elucidated. It is shown that crystallite size decreases with increases in the time needed for the CZTS synthesis stage of the process. The strain in the cell rises in the same way. Generally, the same trend is observed for the process performed at different speeds—for fast processes, the crystallite size is comparatively large and decreases with decreases in the speed as the strain rises. When the temperature rises, the crystallites size decreases, which is in formal contradiction with the results presented by Schurr et al. [18]. However, while the treatments used by Schurr et al. [18] lead to reactions near the equilibrium and it is reasonable to register well-crystallized phases. The time-resolved process requires a shorter time for the equilibrium disposition of the atoms in the structure and formation of the phase that appears in non-equilibrium conditions. As a consequence, the strain in the crystal cell increases. The trends evolved from the RTA features show the better crystallization we can obtain using longer and slower process at comparatively lower temperature limits in the process conditions pointed out in this work. Generally, the order of the structure increases with the increasing duration, speed and temperature of the process, whereas it drops at higher temperatures near 750 °C. The RTA favours the thermal budget of thin absorber films' processing and an appropriate optimization will increase the cost–efficiency ratio.

**Author Contributions:** Conceptualization, T.R., A.M., M.A. and E.M.; Writing—original draft, M.G.; Writing—review & editing, S.S. All authors have read and agreed to the published version of the manuscript.

**Funding:** Part of this work was performed on equipment from the Research Infrastructure "Energy Storage and Hydrogen Energetics" (ESHER), included in the National Roadmap for Research Infrastructure 2017–2023", approved by DCM # 354 from 2017 and granted by the Ministry of Education and Science of Republic Bulgaria. The authors are thankful for Estonian Research Council grant PRG1023, the Operational Program "Science and Education for Smart Growth" through the project "MIRACle"- Grant No BG05M2OP001–1.002–0011-C02 and COST Action 21148 ReNEW PV and Bilateral programs between the Bulgarian Academy of Sciences and Estonian Academy of Sciences—2022–2025.

**Institutional Review Board Statement:** Not applicable.

**Informed Consent Statement:** Not applicable.

**Data Availability Statement:** Not applicable.

**Conflicts of Interest:** The authors declare no conflict of interest.

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
