# Peer review of "Rapid Thermal Processing of Kesterite Thin Films"

_coatings, doi:10.3390/coatings13081449_

Round 1

Reviewer 1 Report

Some suggestions to authors,

 1. The characteristic peaks in the XRD patterns should be marked.

2. Table 2 needs to be modified.

3. Some particles were observed in SEM images. What are they? Meanwhile, how to measure the film thickness? Can you provide the cross-section image?

4. The "-" needs to be deleted in the figure caption in Figs. 12-14. 

Author Response

We appreciate the valuable remarks and suggestions of the esteemed rewiever and present here

our responses and explanations:

Reviewer comments:

  1. The characteristic peaks in the XRD patterns should be marked.

Response: Thanks for the remark. The approach of phase analysis is based on assumption that XRD is not enough itself as instrument of correct determinations and analysis could be completed with Ramman spectroscopy in addition. These are the reasons we describe the data in parallel – just because some formal coincides for a phase are subject of the sequential check for match with resonances by Ramman spectroscopy. The problem is disposed and discussed in lines 107-114

  1. Table 2 needs to be modified.

Response: The suggestion is taken into account and table 2 has been improved.

  1. Some particles were observed in SEM images. What are they? Meanwhile, how to measure the film thickness? Can you provide the cross-section image?

Response: Thanks for remark, the statement is done on personal observations, it is difficult to be presented here and therefore we removed stating of the related parameter. Regarding the cross-section images question, unfortunately we cannot provide them.

  1. The "-" needs to be deleted in the figure caption in Figs. 12-14. 

Response: The descriptions of the figures have been corrected.

Reviewer 2 Report

The manuscript is dedicated to the study rapid thermal processing of kesterite thin films. Time resolved annealing process for CZTS thin films is performed. The peculiarities of phase composition and structure parameters of so formed thin Kesterite layers in dependence of process parameters (as time, speed and temperature) are analysed. A lot of experimental data on films synthesized at various process parameters are presented. However, the data processing is poor, the analysis of the obtained results is poorly carried out, and the manuscript needs the major revision. The following remarks must be taken into account:

1. The abstract should be rewritten by summarizing the problem, the method, the results, and the conclusions.  The first three sentences need to be revised.

2. Abstract, line 23. It is inappropriate to use the abbreviation «CZTS» without decryption. It should be deciphered.

3. Introduction. The introduction section needs to be improved by adding the current advancement and critical review on the progress of kesterite solar cells. The introduction section should be enriched by adding some recent references (reviews), for example,

- Gong, Y., Zhu, Q., Li, B., Wang, S., Duan, B., Lou, L., ... & Xin, H. (2022). Elemental de-mixing-induced epitaxial kesterite/CdS interface enabling 13%-efficiency kesterite solar cells. Nature Energy, 7(10), 966-977,

- Wang, A., He, M., Green, M. A., Sun, K., & Hao, X. (2023). A Critical Review on the Progress of Kesterite Solar Cells: Current Strategies and Insights. Advanced Energy Materials, 13(2), 2203046,

- Yang, S. C., Lin, T. Y., Ochoa, M., Lai, H., Kothandaraman, R., Fu, F., ... & Carron, R. (2023). Efficiency boost of bifacial Cu (In, Ga) Se2 thin-film solar cells for flexible and tandem applications with silver-assisted low-temperature process. Nature Energy, 8(1), 40-51.

or other papers at the discretion of the authors, including in recent years.

4. The aim of the presented work is not clearly stated, which hinders the understanding of the manuscript value. It is necessary to justify the aim of the presented study at the end of the "Introduction" section.

5. The paper presents very accurate XRD results. In section 2, it is necessary to explain the analysis methodology in more detail. Has the XRD analysis been carried out with the reference material having ideal structure? What did the authors do to eliminate the errors of the method?

6. Table 2 needs a more detailed explanation. What are the symbols (dots, diamonds), numbers, signs (plus, plus +minus)? Why for several samples are given «2201+», and for others «9133/-», «●+»,«»?

7. Figure 2 and its description. Here and further according to the XRD results . It is not obviously from figure 2 and its description whether all the peaks are identified? Perhaps the corresponding phase should be indicated in the figure near each peak? It would be better to finish the discussion of the XRD - Figures 2, 4, and then give Figure 3.

8. It is not possible to draw a conclusion about the films thickness from the presented SEM photos. How did the authors evaluate the thickness and continuity of the films?

9. Figure 15. How did the authors build straight lines using three points? Based on what? Here and in other dimensions: error bars should be added.

10. As a result, the authors received a number of contradictory data, which they tried to explain both in section 4 and in the conclusions. It may be advisable to cite the "Discussion" section and describe the mechanism of kesterite formation there, give the reaction equations and explain the data obtained.

11. All abbreviations in the text must be deciphered. Line 143 "RTP". Line 35, the abbreviation "CIGS" appears earlier than it is deciphered.

12. Authors should carefully check the correctness of the References design in accordance with the requirements of the journal. Link 23 is questionable.

13. There are technical errors and typos in the manuscript. For example,

- Line 65: the link needs to be corrected. Similarly lines 278, 326, 349,

- Line 78: check the spelling of the table,

- Line 148: no superscripts,- Table 1 needs to be corrected.

14. All Figures must be issued in a uniform manner, according to the requirements of the journal; the scales must be signed, the font size on the scales must be uniform, easy to read, etc. For example,

- figure12 – scale bar isn’t good for viewing.

- Figures 2, 5a, 7a. The scale caption first indicates the magnitude, then the units of its measurement: «2Theta, Degree».

- Figures 3, 4, 5 (b-d): Scales are not signed,

- Figures 9, 10, 11: “angstrom” must be written correctly.

15. English in the manuscript should be improved.

Author Response

We appreciate the valuable remarks and suggestions of the esteemed rewiever and present here

our responses and explanations:

Reviewer comments:

  1. The abstract should be rewritten by summarizing the problem, the method, the results, and the conclusions.  The first three sentences need to be revised.

Response: We took into account the advice and the Abstract is revised accordingly suggestions:

  1. Abstract, line 23. It is inappropriate to use the abbreviation «CZTS» without decryption. It should be deciphered

Response: The suggestion is taken into account and related remarks are done

  1. The introduction section needs to be improved by adding the current advancement and critical review on the progress of kesterite solar cells. The introduction section should be enriched by adding some recent references:

Response: We are thankful for the advice and the related corrections are made: adding the mentioned works and in addition several more with related discussion

  1. The aim of the presented work is not clearly stated, which hinders the understanding of the manuscript value. It is necessary to justify the aim of the presented study at the end of the "Introduction" section

Response: The advice is accepted and several additions are made, respectively:
The aim of the work is to investigate the features of phase composition, films morphology and defects distribution in results of a rapid thermal annealing (RTA) of electrodeposited CZTS thin precursors films. On the base of the obtained results by XRD and Raman structural analysis are determined the phase composition, contain of the target CZTS, distribution of concomitants and in addition, by assessment of the ratio of intensities of specific Raman signals to   give idea for tendencies in formation of some specific defects (as Cu/Zn substitutions) and approach for management the structural features by process parameters.

  1. The paper presents very accurate XRD results. In section 2, it is necessary to explain the analysis methodology in more detail. Has the XRD analysis been carried out with the reference material having ideal structure? What did the authors do to eliminate the errors of the method?

Response: The advice is accepted and related additions are made in explanation of the methodology:
Qualitative phase analysis was performed on specialized software PDXL Rigaku’s ICDD PDF2 phase research platform using the specifications by International Center for Diffraction Data and EDAX data for chemical composition. The sug-gested phase distribution by XRD is based on detection of three or more main specific re-flects of the phase in compare with the actual data base cards available at the moment.
Answering of the question we count of so described methodology for performance of the XRD analysis and in addition every suggestion for presenting of a phase there was checked personally for compliance with the related  ICDD PDF2 phase card

  1. Table 2 needs a more detailed explanation. What are the symbols (dots, diamonds), numbers, signs (plus, plus +minus)? Why for several samples are given «2201+», and for others «9133/-», «+»,«»?
    Response: We are sorry for the omission and the corrections are made

  1. Figure 2 and its description. Here and further according to the XRD results. It is not obviously from figure 2 and its description whether all the peaks are identified? Perhaps the corresponding phase should be indicated in the figure near each peak? It would be better to finish the discussion of the XRD - Figures 2, 4, and then give Figure 3.

Response: Thanks for the remark. The approach of phase analysis is based on assumption that XRD is not enough itself as instrument of correct determinations and analysis could be completed with Ramman spectroscopy in addition. These are the reasons we describe the data in parallel – just because some formal coincides for a phase are subject of the sequential check for match with resonances by Ramman spectroscopy. The problem is disposed and discussed in lines 107-114

  1. It is not possible to draw a conclusion about the films thickness from the presented SEM How did the authors evaluate the thickness and continuity of the films?

Response: Thanks for remark, the statement is done on personal observations, here is difficult to be presented and it is removed.

  1. Figure 15. How did the authors build straight lines using three points? Based on what? Here and in other dimensions: error bars should be added:

Response: Thanks for the comments. Fig.15 presents ratio of arbitrary intensities of Raman spectra and as far as they are just statistical data it seems difficult to randomize deviations and further, to set error bars. In addition, here we use the figures to present generally, qualitatively the direction of the changes and as it is explained in the section, there are levels definitely above 1 (whish is indicator for well-ordered structure [as is stated in M.Paris, L.Choubrac et.al.]) on one side, and on the other – three points seems obvious to present direction of changes

  1. As a result, the authors received a number of contradictory data, which they tried to explain both in section 4 and in the conclusions. It may be advisable to cite the "Discussion" section and describe the mechanism of Kesterite formation there, give the reaction equations and explain the data obtained.

Response: Thanks for the comments. The discussion, conclusions and abstract are synchronized by sense, that the RTA in the limits of our investigations in all cases (with one exception) leads to formation of well ordered Kesterite (Q > 1) and treatments are favorable for further improvements. In addition, the first suggestion for reaction path of interactions for crystallization of Kesterite is given by R. Schurr et al. and it is cited here, followed by brief comments of important steps:

The reactive annealing is performed in compliance with both suggested reaction path and structural features in [12 R. Schurr]. In conformity of the proposed reaction sequence the synthesis and structure formation of CZTS begin with formation of binaries Cu2S and SnS2 , followed by their interaction to the ternary Cu2SnS3 which at higher temperatures reacts with ZnS and completes as Cu2ZnSnS4 at temperatures higher than 600 °C.

  1. All abbreviations in the text must be deciphered. Line 143 "RTP". Line 35, the abbreviation "CIGS" appears earlier than it is deciphered.

Response: The remark is taken into account and everywhere the abbreviations are used just after deciphering.

  1. Authors should carefully check the correctness of the References design in accordance with the requirements of the journal. Link 23 is questionable.

Response: The references list has been corrected.

  1. There are technical errors and typos in the manuscript.

Response: The text was revised. We fixed all typos and errors in the related places.

  1. All Figures must be issued in a uniform manner, according to the requirements of the journal; the scales must be signed, the font size on the scales must be uniform, easy to read, etc.

Response: The comments are taken into account and errors are edited.

  1. English in the manuscript should be improved.

Response: We did our best to revise and improve the language of the manuscript.

Round 2

Reviewer 2 Report

The manuscript is dedicated to the study rapid thermal processing of kesterite thin films. The manuscript is of interest to the readers, has been improved by the authors and is recommended for publication in the journal "Coatings". 

Minor editing of English language required

Author Response

Thank you!